# Probiogenomics of *Lactobacillus delbrueckii* subsp. *lactis* CIDCA 133: In Silico, In Vitro, and In Vivo Approaches

**DOI:** 10.3390/microorganisms9040829

**Published:** 2021-04-14

**Authors:** Luís Cláudio Lima de Jesus, Mariana Martins Drumond, Flávia Figueira Aburjaile, Thiago de Jesus Sousa, Nina Dias Coelho-Rocha, Rodrigo Profeta, Bertram Brenig, Pamela Mancha-Agresti, Vasco Azevedo

**Affiliations:** 1Laboratório de Genética Celular e Molecular (LGCM), Departamento de Genética, Ecologia e Evolução, Instituto de Ciências Biológicas, Universidade Federal de Minas Gerais (UFMG), Belo Horizonte 31270-901, Brazil; lc.luiis@yahoo.com.br (L.C.L.d.J.); faburjaile@gmail.com (F.F.A.); thiagojsousa@gmail.com (T.d.J.S.); ninadias008@gmail.com (N.D.C.-R.); profeta.biotec@gmail.com (R.P.); 2Centro Federal de Educação Tecnológica de Minas Gerais (CEFET/MG), Departamento de Ciências Biológicas, Belo Horizonte 31421-169, Brazil; mmdrumond@gmail.com; 3Laboratório de Flavivírus, Instituto Oswaldo Cruz, Fundação Oswaldo Cruz, Rio de Janeiro 21040-360, Brazil; 4Institute of Veterinary Medicine, University of Göttingen, D-37077 Göttingen, Germany; bbrenig@gwdg.de; 5Faculdade de Minas-Faminas-BH, Medicina, Belo Horizonte 31744-007, Brazil; p.mancha.agresti@gmail.com

**Keywords:** genomic characterization, phylogenomic, probiotic, *Lactobacillus delbrueckii*, acid and bile tolerance, *Nfkb1* gene expression, surface proteins, bacteriocin

## Abstract

*Lactobacillus delbrueckii* subsp. *lactis* CIDCA 133 (CIDCA 133) has been reported as a potential probiotic strain, presenting immunomodulatory properties. This study investigated the possible genes and molecular mechanism involved with a probiotic profile of CIDCA 133 through a genomic approach associated with in vitro and in vivo analysis. Genomic analysis corroborates the species identification carried out by the classical microbiological method. Phenotypic assays demonstrated that the CIDCA 133 strain could survive acidic, osmotic, and thermic stresses. In addition, this strain shows antibacterial activity against *Salmonella* Typhimurium and presents immunostimulatory properties capable of upregulating anti-inflammatory cytokines *Il10* and *Tgfb1* gene expression through inhibition of *Nfkb1* gene expression. These reported effects can be associated with secreted, membrane/exposed to the surface and cytoplasmic proteins, and bacteriocins-encoding genes predicted in silico. Furthermore, our results showed the genes and the possible mechanisms used by CIDCA 133 to produce their beneficial host effects and highlight its use as a probiotic microorganism.

## 1. Introduction

Lactobacillus is a highly diverse taxonomic group of Gram-positive microorganisms, rod or coccobacilli-shaped, members of lactic acid bacteria (LAB), facultatively anaerobic [1,2], and able to produce lactic acid as the primary metabolic end product of carbohydrate fermentation [2,3]. These microorganisms can be found and isolated from different ecological niches (e.g., vegetables, fermented products, gastrointestinal and vaginal tracts of humans and animals) where there is a high carbohydrate availability [4].

Many *Lactobacillus* strains have a probiotic profile and, thus, present functional characteristics beneficial to the host, such as their immunomodulatory and anti-inflammatory properties [5,6], and its effectiveness on the treatment of Crohn’s disease and ulcerative colitis [7,8], intestinal mucositis [9,10], and enteric infections [11,12]. However, it should be emphasized that the beneficial effects of probiotics on the host are strain-dependent [13,14] and cannot be generalized.

The *Lactobacillus* strains described as probiotics, with many potential benefits attributed to the host health, and general commercial and biotechnological potential, are now being studied in the genomic field [15,16,17]. The *Lactobacillus* genome analysis has contributed to more detailed characterization in terms of the identification and function of gene products and possible molecular mechanisms related to the probiotic effects attributed to these bacteria [18], as well as their animal and human consumption-related safety [16,19]. However, unlike other LAB species, few studies have focussed on *Lactobacillus delbrueckii* probiogenomics, and the few genomic data are mainly obtained from bulgaricus subspecies [20,21].

*Lactobacillus delbrueckii* subsp. *lactis* CIDCA 133 (CIDCA 133) is a new potential probiotic strain of *Lactobacillus delbrueckii* subsp. *lactis* species isolated from raw cow’s milk [22]. According to previous in vitro studies, this strain has shown the ability to tolerate high concentrations of bile salts [23] and antagonistic action against the pathogenic enterohemorrhagic *Escherichia coli* (EHEC) [24] and bacteria causing food contamination, such as *Pseudomonas aeruginosa* [22]. Furthermore, this probiotic strain modulated cell response in both monocyte-derived dendritic cells and murine macrophages (RAW 264.7 cells) infected with *Bacillus cereus* [25] and *Citrobacter rodentium* [26] through enhancing the TNF-α and ROS production, respectively. This probiotic strain was also able to survive and grow in the presence of enterocyte-derived antimicrobial molecules, such as β-defensins [27,28]. Additionally, it was also reported that the administration of fermented milk by CIDCA 133 to BALB/c mice prevented the inflammatory response and histopathological damage caused to the intestinal mucosa after 5-Fluorouracil (300 mg/Kg) chemotherapy administration [10].

Although promising results have been previously obtained in pre-clinical studies regarding this strain, there is little information about the genetic factors related to its protective action mechanisms; therefore, this work aims to characterize the *L. delbrueckii* CIDCA 133 strain through a probiogenomics approach. The genomic data (in silico) will allow the potential molecular mechanisms involved with the probiotic properties and immunomodulatory capacity for this strain to be known and reported in vitro and in vivo studies.

## 2. Materials and Methods

### 2.1. Bacteria Strain and Growth Conditions

*Lactobacillus delbrueckii* subsp. *lactis* CIDCA 133 belongs to the culture collection of the Centro de Investigación y Desarrollo en Criotecnología de Alimentos (CIDCA) of the Universidad Nacional de La Plata, Argentina. This strain was deposited at the Bacteria Collection from Environment and Health (CBAS) of the Oswaldo Cruz Foundation (FIOCRUZ) (Accession number: CBAS 815). CIDCA 133 was cultivated in de Man, Rogosa, and Sharpe (MRS) broth (Kasvi, São José dos Pinhais, Brazil) for 16 h at 37 °C.

### 2.2. CIDCA 133 Identification by MALDI-TOF Biotyper^®^

After growth, CIDCA 133 was plated using a sterile plastic loop on MRS agar plates (São José dos Pinhais, Brazil) and incubated at 37 °C for 48 h. he colonies’ identification using the MALDI-TOF Biotyper^®^ Mass Spectrometry (Brukker Daltoniks, Billerica, MA, USA) was performed according to the manufacturer’s instructions.

### 2.3. Genomic and Plasmid DNA Extraction

CIDCA 133 genomic DNA extraction was performed through mechanical lysis followed by purification with a phenol solution (phenol:chloroform:isoamyl alcohol 25:24:1, *v*/*v*, respectively), precipitation with ethanol 70% and sodium acetate 3 M, and suspension in DNase and RNase-free water, according to the protocol established by Sachinandan et al. [29]. According to the manufacturer’s instructions, the plasmid extraction was performed using the Pure Link™ Quick Plasmid Miniprep Kit (Invitrogen, Carlsbad, CA, USA).

### 2.4. Genome Sequencing, Assembly, and Annotation

CIDCA 133 whole genome sequence was performed using the HiSeq 2500 platform (Illumina, San Diego, CA, USA), paired-end libraries (2 × 150 bp). The quality assessment of the reads was performed using FastQC [30]. De novo genome assembly was performed using the Edena assembler (v. 3.13) [31]. The assembly quality was verified using QUAST (Quality assessment tool) [32]. The contigs were ordered and oriented through the *CONTIGUATOR* (v. 2.74) [33], using the whole genome of *L. delbrueckii* subsp. *bulgaricus* ND02 (RefSeq: NC_014727.1) as a reference. The remaining gaps were closed using GapBLASTER (v. 1.1.2) [34], GenomeFinisher (v.1.4) [35] and QIAGEN CLC Genomics Workbench 20 (v. 20.0.4) [36].

The protein encoding-ORFs were automatically annotated using the Prokaryotic Genome Annotation Pipeline (PGAP) from the National Center for Biotechnology (NCBI) [37]. The genome and plasmid sequences were deposited in the NCBI (Access Number: CP065513 and CP065514, respectively).

### 2.5. In Silico Analysis

#### 2.5.1. Plasmid Identification

The presence of plasmids was searched using the PlasmidFinder (https://cge.cbs.dtu.dk/services/PlasmidFinder/) (accessed on 15 December 2020) [38]. The circular map of the plasmid was obtained by SnapGene^®^ software (v. 5.1.3.1).

#### 2.5.2. Phylogenomic Analysis

For comparative genomic analysis, 26 complete genomes of *Lactobacillus delbrueckii* strains from the NCBI database were used (Table 1). The taxonomic analysis to compare whether or not the strains belonged to the same species was carried out by calculating the Average Nucleotide Identity (ANI) by Blast (ANIb) performed within the JSpecies Web Server [39]. ANIb values were visualized as a heatmap. Genomes with ANI > 95% were considered the same species.

#### 2.5.3. Subcellular Localization of CIDCA 133 Proteins

The prediction of subcellular localization of CIDCA 133 proteins was performed using the SurfG+ software, which classifies proteins based on the presence (secreted proteins) or absence (cytoplasmic proteins) of a signal peptide, transmembrane helices (membrane proteins), and signal retention (proteins that are covalently or transiently bound to the cell wall) [40].

#### 2.5.4. Functional Annotation of CIDCA 133 Proteins

For functional characterization, the protein sequences predicted in the CIDCA 133 genome were submitted to the GO FEAT (Gene Ontology Functional Enrichment Annotation Tool) (http://computationalbiology.ufpa.br/gofeat/) (accessed on 8 January 2021) [41].

#### 2.5.5. Cell Adhesion-Related Genes

The CIDCA 133 genes’ prediction involved in the adhesion mechanisms was evaluated through Vaxign (Vaccine Design) (v.2beta) (http://www.violinet.org/vaxign/) (accessed on 17 January 2021) [42]. A score of 0.6 was selected as the criteria to analyze which CIDCA 133 proteins (sub-located in the membrane, surface-exposed (PSE), or secreted) have a high adhesion capacity.

#### 2.5.6. Proteolytic Activity and Stress Tolerance-Related Genes

The CIDCA 133 stress tolerance (acid, bile, thermal, and osmotic) and proteolytic system-related genes’ prediction were based on literature data revision [43,44,45,46,47,48] for previously reported genes identified in probiotic bacteria involved with the above processes.

#### 2.5.7. Metabolic and Symbiotic Islands Prediction

The Metabolic (MI) and Symbiotic (SI) Islands prediction in the CIDCA 133 genome was performed with GIPSy software (Genomic Island Prediction Software) (v.1.1.2) [49], using the *Lacticaseibacillus rhamnosus* GG (*L. rhamnosus*) genome (NZ_CP031290.1) as subjects. The Genomic Islands (GEIs) map was visualized using BRIG (BLAST Ring Image Generator) software (v. 0.95) [50].

#### 2.5.8. Bacteriocins Prediction

Gene’s prediction related to bacteriocins synthesis was performed by BAGEL4 (BActeriocin GEnome mining tooL) (http://bagel4.molgenrug.nl/index.php) (accessed on 22 December 2020) [51].

#### 2.5.9. Protein–Protein Interactions Prediction

For the potential biological functions of CIDCA 133 on human immunology, the prediction of interactions between CIDCA 133 and human proteins was carried out. The human protein sequence was mapped to KEGG pathways (toll-like receptor 2/4 nuclear factor κappa B (TLR2/4-NF-κB) pathway) and obtained from UniProt (UP000005640) (Appendix A). The CIDCA 133 proteins with a high likelihood of adherence predicted by Vaxign (>0.6 scores) were used. The protein–protein interaction was performed in InterSPPI [52]. The resulting interactions were filtered according to the 0.9765 score prediction (specificity of 0.99). The graphical interaction results were achieved by Cytoscape software [53].

### 2.6. In Vitro Analysis

#### 2.6.1. Simulated Gastric Juice and Heat Stress Tolerance

The CIDCA 133 tolerance to acidic gastric juice simulated with pepsin solution (pH 3.0) was performed according to Singhal et al. [54]. Briefly, 3 g/L of pepsin (Sigma–Aldrich, St. Louis, MO, USA) was diluted in 0.5% sterile NaCl (pH 3.0) (Vetec, Rio de Janeiro, Brazil). Subsequently, the cell pellet (10^8^ CFU/mL) was washed twice with sterile and cold PBS 0.1 M (pH 7.0) and suspended with 400 µL of sterile NaCl 0.5% (pH 7.0). One hundred microliters of the culture was inoculated in 900 µL of the pepsin solution (pH 3.0) and incubated at 37 °C for 4 h with shaking (200 rpm) in a shaker (Labnet, Edison, NJ, USA)

For heat stress, the CIDCA 133 culture (10^8^ CFU/mL) was centrifuged (5000 rpm for 10 min at 4 °C), washed twice with sterile and cold PBS 0.1 M (pH 7.0), suspend with 1 mL of MRS broth, and incubated for 30 min to 65 °C (a temperature of the simulated pasteurization process) [55]. As a control, 1 mL of CIDCA 133 was not submitted to heat stress.

Then, 100 µL of each sample was collected after 0, 2, and 4 h (acid stress) and 30 min (heat stress) of incubation, and serially diluted (1:10) (acid stress: 10^−8^; heat stress: 10^−7^) in sterile-cold PBS 0.1 M (pH 7.0), plated on MRS agar (Kasvi, São José dos Pinhais, Brazil) and incubated at 37 °C for 48 h. The number of viable bacteria was determined by counting colony-forming units (CFU/mL) after incubation period.

#### 2.6.2. Osmotic Stress Tolerance

For CIDCA 133 ability to tolerate different concentrations of sodium chloride (NaCl), 150 µL of the culture was inoculated in 15 mL of MRS broth containing different concentrations of NaCl (1%, 2%, 3%, 4%, and 5%) [56]. As a control, 150 µL of CIDCA 133 was inoculated in 15 mL of MRS broth without NaCl supplementation. After 24 h of growth at 37 °C, the samples’ absorbance was measured at O.D._600_ nm.

#### 2.6.3. Antibacterial Activity

For this analysis, the indicator strains *Shigella sonnei* ATCC^®^ 9290, *Salmonella enterica* serovar Typhimurium ATCC^®^ 29630, *Enterococcus faecalis* ATCC^®^ 19433, *Listeria monocytogenes* ATCC^®^ 15313 were obtained from the American Type Culture Collection (ATCC) (Manassas, Virginia, EUA). *Lactobacillus delbrueckii* CNRZ327 and *Lacticaseibacillus paracasei* BL23 (*L. paracasei* BL23) belongs to the culture collection of the Institute Nacional de la Recherche Agronomique (INRA, Jouy-en-Josas, France). These strains were cultivated in MRS broth (Kasvi, São José dos Pinhais, Brazil) or BHI (Brain Heart Infusion) (Sigma–Aldrich, St. Louis, MO, USA) at 37 °C for 24 h.

Antibacterial activity of CIDCA 133 against these indicator strains was evaluated using CIDCA 133 cells-free supernatant (CFS), according to the method described by Somashekaraiah et al. [57], with some modifications. For this purpose, 100 mL of CIDCA 133 culture grown in MRS broth at 37 °C for 24 h was centrifuged (5000 rpm for 15 min at 4 °C). Part of the cell-free supernatants (CFS) was kept with their initial acid pH. Another was neutralized (nCFS) (pH 6.5) with 1.0 M NaOH (Vetec, Rio de Janeiro, Brazil). The CFS and nCFS aliquots were sterilized through a 0.22 μm filter (Kasvi, São José dos Pinhais, Brazil). Then, 200 μL of the indicator strains, previously grown in BHI broth at 37 °C for 24 h, was inoculated in 2 mL of the CIDCA 133 supernatant (CFS or nCFS). As a control, the indicator strains were grown in MRS broth. After 24 h incubation at 37 °C, the O.D._600_ nm was measured.

### 2.7. In Vivo Analysis

#### 2.7.1. Gene Expression of Cytokines in Mice Ileum

The experiments were conducted on male BALB/c mice (weight 25–30 g, six weeks old) obtained from Centro de Bioterismo (CEBIO) of the Institute of Biological Sciences at the Federal University of Minas Gerais (UFMG). The animals were kept in polycarbonate-ventilated cages under controlled conditions: temperature around 21 ± 2 °C with a 12-h light/dark cycle, and ad libitum access to water and standard chow diet 24 h before experiments. All procedures followed the Brazilian College of Animal Experimentation (COBEA), and the Local Animal Experimental Ethics Committee (CEUA-UFMG) approved the project (Protocol n° 112/2020).

#### 2.7.2. CIDCA 133 Administration

Mice were randomized into two experimental groups (*n* = 6 animals per group): I- NC (negative control) and II- CIDCA 133. These groups were administered by continuous feeding with 100 mL/cage of MRS broth (CTL group) or CIDCA 133 (5 × 10^7^ CFU/mL) for 13 consecutive days. After the experimentation period, the animals were euthanized by a single intraperitoneal injection of anesthetic overdose (30 mg/kg of xylazine and 300 mg/kg of ketamine mixture) (Ceva, São Paulo, Brazil) and samples of the intestine (ileum section) were collected and stored in RNAlater^®^ solution (Invitrogen, Carlsbad, CA, USA).

#### 2.7.3. RNA Extraction and Quantitative Polymerase Chain Reaction (qPCR)

According to the manufacturer’s instructions, the total RNA of ileum sections (~20 mg) was obtained using the RNeasy Mini Kit (QIAGEN, Hilden, Germany). The RNA quality and concentration were evaluated on 1.5% agarose gel electrophoresis and through the NanoDrop^®^ 2000 spectrophotometer (Thermo Scientific, Waltham, MA, USA), respectively. Residual DNA was digested with DNAse I from the TURBO DNA-free™ Kit (Invitrogen, Carlsbad, CA, USA), following the manufacturer’s instructions. The complementary deoxyribonucleic acid (cDNA) synthesis was produced with the Applied Biosystems™ High-Capacity cDNA Reverse Transcription kit (ThermoFisher, Waltham, MA, USA), according to the manufacturer’s instructions.

Quantitative PCR (qPCR) was performed using the PowerUp™ SYBR^®^ Green Master Mix (ThermoFisher, Waltham, MA, USA) and the genes-specific primers for *Tlr2*, *Tlr4*, *Myd88*, *Nfkb1*, *Tnf*, *Il1b*, *Il6*, *Il12*, *Il10*, *Il17a*, *Tgfb1*, and *Muc2* (Table 2). Amplification reactions were performed on the Applied Biosystems 7900HT Fast Real-Time PCR System under the following conditions: initial denaturation at 95 °C for 10 min, 95 °C for 15 seg, annealing/extension at 60 °C for 1 min, 40 cycles followed by a dissociation stage for recording the melting curve. The expression of target genes was analyzed by the 2^−ΔΔCt^ method using housekeeping genes encoding β-actin (*actb*) and GAPDH (*gapdh*) as endogenous references.

### 2.8. Statistical Analysis

The experiments were done in triplicate (gastric juice, osmotic and thermal stress tolerance, bacterial antagonism) or duplicate (qPCR analysis). The results were presented as mean and standard deviation (SD). Statistical differences between the two groups were performed by the Student’s *t*-test (thermal stress tolerance, qPCR, and bacterial antagonism analysis). Stress experiments (gastric juice and osmotic stress tolerance) were performed by analyzing variance (ANOVA) followed by Tukey’s post hoc test. All data were analyzed using the GraphPad Prism 8.0 software, and a *p*-value < 0.05 was considered significant.

## 3. Results

### 3.1. L. delbrueckii CIDCA 133 General Genomic Features

Genome sequencing of *L. delbrueckii CIDCA 133* strain revealed a single circular chromosomal DNA of 2,127,785 bp, with a GC% content of 49.57%, 27 rRNA, 98 tRNA, 153 pseudogenes, 2132 genes, and a total of 2004 protein-coding sequences (CDS). Additionally, the presence of one plasmid sequence was detected in CIDCA 133 (Figure 1). This plasmid had 6224 bp, a GC content of 44.67%, and six CDSs.

### 3.2. Gene Ontology (GO) Annotation

A total of 1590 genes of CIDCA 133 exhibited results in the GO FEAT platform’s functional annotation. The GO terms were represented in three categories: molecular function (50.94% hits), biological process (27.06% hits), and cell component (22% hits) (Figure 2A).

The cellular component category contained GO terminologies involved in membrane function (integral components of the membrane, plasm membrane) and cytoplasmic function (ribosome), among others (Figure 2B). For molecular function, it was identified that the main GO terminologies functions referred to protein binding (DNA, ATP, and metal-binding) and catalytic activity (ATPase and hydrolase activity), among others (Figure 2C).

For the biological process category, the most representative GO terms were translation, transmembrane transport, DNA repair, and carbohydrate metabolism (Figure 2D).

### 3.3. Species Identification

The CIDCA 133 identification by MALDI-TOF Biotyper^®^ classified this strain as belonging to the *L. delbrueckii* species, but with a certain degree of uncertainty (score < 2.2). However, pairwise comparisons of the Average Nucleotide Identity based on BLAST (ANIb) indicate that CIDCA 133 genome presented an identity threshold > 97% with 26 *L. delbrueckii* genomes (Figure 3), consistent with their identification as members of the same species.

ANIb distance between the strains indicated the formation of two main clades: one represented by strains of *L. delbrueckii* subsp. *delbrueckii*, *L. delbrueckii* subsp. *jakobsenii* and *L. delbrueckii* subsp. *lactis* (red region upper right), and the other included strains of *L. delbrueckii* subsp. *bulgaricus* (red region, lower left). This phylogenomic analysis shows that *L. delbrueckii* CIDCA 133 is closely related to the clade of the *L. delbrueckii* subsp. *lactis* species (ANIb > 98%).

### 3.4. CIDCA 133 Tolerates Acid, Osmotic and Thermal Stresses

Genes coding for proteins involved in acid, thermal, osmotic and bile salt resistance were identified in CIDCA 133 genome. These genes encode proteins as ornithine decarboxylase, F0F1-ATP synthase (acid stress), Na (+)/H (+) antiporter NhaC, aquaporin family protein (osmotic stress), choloylglycine hydrolase, S-ribosylhomocysteine lyase (bile salt stress), chaperones (GroEL, DnaK) (heat stress), among others (Table 3).

Additionally, the capacity of CIDCA 133 to tolerate these stressors agents was evaluated. For acid stress, it was observed that compared to the initial time (0 h), the viability of CIDCA 133 decreased after 2 h and 4 h in contact with artificial gastric juice, but the strain continued to maintain a high survival rate: 77.7% (2 h) and 67.4% (4 h), thus being able to grow after acid pH challenge (Figure 4A).

For osmotic stress, no change in the growth of CIDCA 133 was observed in the presence of 1%, 2%, and 3% NaCl. The strain showed a growth rate of 99.5%, 98.1%, and 87.6%, respectively. These results are like bacteria that were not submitted to osmotic stress (NaCl 0%; 100% in the growth rate). However, when the NaCl concentration was increased to 4% and 5%, the strain had a growth rate of 48.1% and 31.7%, respectively, revealing that these high concentrations of NaCl reduce the CIDCA 133 growth (Figure 4B).

After heat stress, it was observed that CIDCA 133 presented 63.75% of viability, revealing that the strain can tolerate high temperature (Figure 4C).

### 3.5. Proteolytic System, Symbiotic, and Metabolic Genomic Islands

Based on data from the literature, through manual inspection of the CIDCA 133 genome annotation, it was possible to identify CDS possibly related to the strain proteolytic activity. CIDCA 133 genome encodes genes related to cell-wall bound proteinase (*PrtB*, *PrtM*), different classes of peptidases (*pepN*, *pepC*, *pepV*, *pepT*, *pepO*, *pepX*), and peptide transporters (*oppA*, *oppC*, *dppB*, *dppE*) (Appendix A).

Additionally, twelve genomic islands (GEIs) were identified: seven symbiotic (SI) and five metabolic islands (MI), respectively (Figure 5). All CDS of CIDCA 133 GEIs are described in Appendix A.

### 3.6. Putative Bacteriocins and Antibacterial Activity

In CIDCA 133 genome, the BAGEL4 web server predicted three bacteriocins belonging to class III: two helveticin-J (Figure 6A,C) (330 and 331 amino acids, respectively) and enterolysin-A (Figure 6B) (269 amino acids).

Additionally, to determine whether *L. delbrueckii* CIDCA 133 exhibited antibacterial activity, a bacterial inhibition assay was performed based on inhibitory compounds present in its culture supernatant. The bacterial culture supernatant (CFS) had an approximate pH of 3.8 after 24 h of growth and was able to inhibit the growth of pathogens, such as *L. monocytogenes* (90.9% ± 1.70), *E. faecalis* (88.4% ± 7.5), *S. sonnei* (91.6% ± 2.6), and *S. enterica* Typhimurium (84% ± 11.1). In addition, the CIDCA 133 supernatant effect was evaluated in other *Lactobacillus* species, such as *L. delbrueckii* CNRZ327 and *L. paracasei* BL23, in which it was possible to observe an inhibition rate against these bacteria of 77.7% ± 4.9 and 88.8% ± 0.14, respectively (Figure 6D). After the neutralization of supernatant (nCFS) (pH = 6.5), it was possible to observe a reduction in the inhibition rate of *L. monocytogenes* (35.9% ± 1.12; *p =* 0.0007), *E. faecalis* (30.95% ± 0.3; *p =* 0.0086), *S. sonnei* (39.7% ± 6.6; *p =* 0.0094), *S. enterica* Typhimurium (34.71% ± 3.3; *p =* 0.0267), *L. paracasei* BL23 (29.4% ± 4.1; *p =* 0.0024), and *L. delbrueckii* CNRZ327 (42.41% ± 8.7; *p <* 0.05) (Figure 6D).

### 3.7. Cell Adhesion-Related Genes

Using SurfG+ software, it was found that 1606 proteins of CIDCA 133 are cytoplasmic (CYT), 312 membrane (ME), 156 protein surfaces exposed (PSE), and 58 secreted (SE) (Figure 7A; Appendix A). According to the Vaxign web server, 16 of the predicted proteins sub located on the membrane, 48 PSE and 38 secreted had high cell adhesion probability (Figure 7B; Appendix A). These proteins-encoding genes include the SLAP domain-containing protein (*SLAP*)*,* peptidase S8 (*PrtB*)*,* MucBP domain protein (*MucB*)*,* aggregation promoting factor (*Apf*)*,* lipoteichoic acid synthase family protein (*LtaS*)*,* trypsin-like serine protease (*HtrA*), among others.

### 3.8. Protein-Protein Interaction

The CIDCA 133 and human protein–protein interaction by InterSPPI predicted 74 interactions (Appendix A). The nuclear factor NF-κB p105 subunit (NFKB1) was the most frequent interacting human protein. On the other hand, the PrtB protein (a cell surface proteinase) was the most frequent interacting CIDCA 133 protein. CIDCA 133 proteins also interacted with other human proteins involved with the TLR/NF-κB signaling pathway activation, such as TLR4, IRAK4, IRAK1, TRAF6, TAB2, TAK1, IKKB, RELA, and NFKBIA (Figure 7C).

### 3.9. CIDCA 133 Influences on Intestinal Mucosa Immune System

Consumption of CIDCA 133 was also reported to modulate ileal expression of cytokines genes in mice. After oral CIDCA 133 administration, it was possible to observe a downregulation in the mRNA expression of inflammatory cytokines *Tnf* (0.79 ± 0.12), *Il6* (0.67 ± 0.11), *Il12* (0.42 ± 0.16), *Il1b* (0.56 ± 0.14), and *Il17a* (0.31 ± 0.25) when compared to those exhibited in the NC group: *Tnf* (1.00 ± 0.18; *p =* 0.0406), *Il6* (1.00 ± 0.20; *p =* 0.0064), *Il12* (1.00 ± 0.19; *p* = 0.0004), *Il1b* (1.00 ± 0.18; *p =* 0.0011), and *Il17a* (1.00 ± 0.30; *p =* 0.0017) (Figure 8).

On the other hand, the mRNA expression of *Muc2* (1.48 ± 0.19), *Tlr2* (2.19 ± 0.59), *Tlr4* (1.90 ± 0.41), *Myd88* (4.22 ± 0.57), *Nfkb1* (0.44 ± 0.15), *Tgfb1* (1.56 ± 0.48), and *Il10* (1.63 ± 0.18) were upregulated after oral administration of CIDCA 133 in relation to the NC group: *Muc2* (1.00 ± 0.12; *p =* 0.0010), *Tlr2* (1.00 ± 0.20; *p =* 0.0028), *Tlr4* (1.00 ± 0.30; *p =* 0.0052), *Myd88* (1.00 ± 0.67; *p <* 0.0001), *Nfkb1* (1.00 ± 0.19; *p =* 0.0003), *Tgfb1* (1.00 ± 0.27; *p =* 0.0325), and *Il10* (1.00 ± 0.15; *p =* 0.0002) (Figure 8).

## 4. Discussion

*Lactobacillus* strains have functional characteristics beneficial to the host, such as an anti-inflammatory effect and resistance and adaptation mechanisms to the GIT conditions. These features lead these microorganisms to have high relevance in the biotechnological and industrial food sector [5,63] for their use as a probiotic supplement. Due to these properties, specific mechanisms of action of these microorganisms have been elucidated through omics investigations [18].

*L. delbrueckii* CIDCA 133 has emerged as a potential probiotic strain [10,22,25]. Based on its beneficial effects, species identification, gene product function, and potential molecular mechanisms associated with these strain’s probiotic effects were investigated in this work through a genome and phenotype-scale analysis.

CIDCA 133 had its identification performed by classical microbiological methods. Both MALDI-TOF Biotyper^®^ and Average Nucleotide Identity (ANI) analysis supported this classification, which showed that this strain presents a high similarity with the others belonging to the *L. delbrueckii* subsp. *lactis* species [64]. The CIDCA 133 genome had about 2.2 Mb and 2004 protein-coding sequences. In addition, this strain had one plasmid sequence (6224 bp). According to Lee et al. [65], the presence of plasmids in *L. delbrueckii* strains is rare, unlike other Lactic Acid Bacteria (LAB) species. The low number of plasmid sequences of this species deposited in the NCBI corroborates this fact, with only four plasmid sequences deposited (Access Number: CP002342.1; CP018612.1; CP018613.1 and CP029251.1).

Probiotic microorganisms must resist stress in both product matrices and during their passage through the GIT to produce many beneficial effects on the host’s health. CIDCA 133 harbored many genes encoded for stress-related proteins, such as a two-component system sensor, F0F1 ATP synthase, ornithine decarboxylase, phosphopyruvate hydratase, and choloylglycine hydrolase. These proteins respond to specific stress stimulus and generate a broad range of response results [45]. Furthermore, these genetic factors can be associated with CIDCA 133 survival capacity to simulated gastric juice reported in this study and to the data found by Kociubinski et al. [23], which demonstrated for the first time the ability of CIDCA 133 to resist bile salt (0.1% and 0.5%). These genes were also previously shown to be involved with the capacity of *L. rhamnosus* [66], *Limosilactobacillus reuteri* (*L. reuteri*) [67], and *L. helveticus* [47] strain to survive to pH 3 and 0.3% bile salt for 2–3 h.

CIDCA 133 also carried genes, such as Na+/H+ antiporter (*NhaC*), S-ribosylhomocysteine lyase (*luXs*)*,* aquaporin family protein (*glpF*)*,* and heat shock proteins (*DnaK, DnaJ, GroEL*), which may be related to its ability to tolerate different concentrations (1%–3%) of sodium chloride (osmotic stress) and heat stress, respectively. These findings corroborate with other studies that demonstrated that different *Lactobacillus* species could tolerate different NaCl concentrations, such as *Lactiplantibacillus plantarum* (*L. plantarum*) [68] that tolerates up to 5% of NaCl, *L. paracasei* [56] and *L. delbrueckii* subsp. *bulgaricus* [69], which tolerated up to 4% NaCl.

Sodium chloride is generally used in the fermented food industry, such as cheese [70]. However, varying concentrations of sodium chloride present in these products and the high temperature used for their production can compromise probiotic bacteria’s viability and activity [71,72]. Therefore, the ability of CIDCA 133 to resist acid, bile, different concentrations of NaCl (1–3%), and pasteurization temperature allows this strain to perform better at its health-promoting site of action and makes it promising for application in the food sector for the development of dairy fermented products with functional characteristics.

When consumed, probiotic bacteria must also have the ability to interact with intestinal epithelial cells, which is a crucial factor for their interaction activation with the host [73,74]. Several studies have demonstrated the involvement of extracellular and surface-bound proteins identified in the bacteria/host interaction, leading to biological processes, such as cell adhesion, competitive exclusion of pathogens, and mucosal immune regulation. These proteins include SlpB, slpE, htrA4, and hsdM3 of *P. freudenreichii* CIRM-BIA 129 and CIRM-BIA 121 [75,76,77], and SlpA of *L. acidophilus* [78] and *L. helveticus* MIMLh5 [79], among others.

In the CIDCA 133 genome, 312 membrane proteins, 58 secreted and 156 surfaces exposed (PSE) were predicted. Of these, 102 were identified with a high probability of cell adhesion, such as the SLAP domain-containing protein, MucBP domain-containing protein, lipoteichoic acid synthase family protein, proteinase PrtB, and aggregation promoting factor. These proteins can be involved in the protective effects of CIDCA 133 against *Bacillus cereus* [25] and *Citrobacter rodentium* [26] infection. This bacteria stimulated immune cell responses (macrophages and dendritic cells derived from human monocytes) infected with these pathogens to reduce the infection by producing co-stimulatory and effector molecules (TNF-α, IL-6, IL-8, and iNOS).

*Lactobacillus* strains can modulate the host’s immune response through their interaction with intestinal epithelial cells [80] mainly conducted by toll-like receptors (TLRs), which when activated can stimulate the activation of signaling pathways, such as the nuclear factor κappa B (NF-κB) and mitogen-activated protein kinase (MAPK), with subsequent production of cytokines [81].

The immunostimulatory capacity of CIDCA 133 in vivo was evaluated in this work. It was possible to observe an increase in the gene expression of *Tlr2*, *Tlr4* and *Myd88* after CIDCA 133 strain consumption. These findings are supported by other studies, which observed that administration of the probiotics *Lacticaseibacillus casei* (*L. casei*) and *Saccharomyces boulardii* could also stimulate the mucosal immune system of healthy mice and broilers, respectively, by increasing gene expression of *Tlr2*, *Tlr4,* and *Myd88* [82,83].

The NF-κB pathway leads to the upregulation of pro-inflammatory genes that, if not controlled at homeostatic levels, can lead to the onset and progression of inflammatory bowel diseases (IBDs) [84,85,86]. Several probiotics can downregulate the expression of pro-inflammatory cytokines. *L. acidophilus* was able to decrease the intestinal damage caused by 5-Fluorouracil (5-FU) (450 mg/kg) by inhibiting the signaling of the NF-κB pathway and observing low levels of pro-inflammatory cytokines TNF-α and IL-1β [87]. *L. gasseri* 4M13 inhibited the release of inflammatory mediators, such as TNF-α, IL-6, IL-1β, and induced IL-10, in LPS-stimulated RAW 264.7 macrophages [88]. In addition, *L. helveticus* SBT2171 induces A20 gene expression for inhibiting the activation of NF-κB/MAPKs and IL-6 and IL-1β production in macrophages cell [89].

Knowing the reduction in pro-inflammatory cytokines expression is also reported as a positive effect of probiotic microorganisms [82,83], in this work, a reduction in pro-inflammatory (*Tnf, Il6, Il12, Il17a,* and *Il1b*) and an increase in anti-inflammatory (*Tgfb1* and *Il10*) cytokines gene expression in health mice was observed after oral CIDCA 133 administration. This modulation can be related to the downregulation of *Nfkb1* gene expression. This result is following the InterSPPI prediction since the nuclear factor NF-κB p105 subunit (NFKB1) was the most frequent interacting human protein with CIDCA 133 proteins, suggesting these proteins are possibly involved with its immunomodulatory property. However, other studies must be performed to validate this finding, such as CIDCA 133 knockout genes or heterologous production of these proteins, and their phenotypic evaluation on inflammation models.

Some studies have also demonstrated that epithelial activation of TLR2/TLR4 is associated with the development and maturation of mucus-producing goblet cells [90,91]. This finding supports the results reported in this work, in which it was observed that oral administration of CIDCA 133 also increased the gene expression of the MUC2 protein (*mucin 2*), one of the main components of the intestinal mucus layer.

Based on these findings, it can also be inferred that the modulation of the epithelial barrier markers and immune system to an anti-inflammatory profile by CIDCA 133 in healthy mice can be associated with its protective effect against intestinal mucosa damage caused by 5-FU chemotherapy [10]. Thus, this property further highlights the anti-inflammatory effect that the CIDCA 133 strain can exert on the host.

The commensal and probiotic bacteria must also act in symbiosis with the host to promote its beneficial effects. The host provides a stable habitat for these microorganisms while providing them with beneficial nutrients [92,93]. In this context, the presence of five metabolic islands (MI), seven symbiotic Islands (SI), and genes related to proteolytic activity in the CIDCA 133 genome (e.g., *OppA, pepC pepI, pepA, PrtB*) highlights the ability of this strain to capture and metabolize dairy proteins during the fermentation process. An organized proteolytic system has also been identified in other *Lactobacillus* species, such as *L. reuteri* [94,95], *L. helveticus* [96], and *Lactiplantibacillus pentosus* (*L. pentosus*) [97].

The proteolytic activity of probiotic bacteria during the fermentation process is much responsible for bioactive peptides production [98] and other compounds, such as vitamins [99] and Short Chain Fatty Acids (SCFA) [100,101], which, besides improving the sensory characteristics of dairy products [102,103], promote beneficial effects to the host due to its antioxidant and immunomodulatory activity. The beneficial effects of fermented formulations by probiotic bacteria, such as *L. rhamnosus* GG [104], *L. delbrueckii* CNRZ327 [105], *L. plantarum* [11], *L. paracasei* BL23, and *P. freudenreichii* 138 [9], has been reported due to their effectiveness for preventing enteric infection, and the intestinal inflammation and histological damage in murine models of colitis and mucositis disease. The beneficial effects of dairy fermented product by CIDCA 133 were previously reported in a murine model of mucositis [10], evidencing, therefore, the intrinsic and healthy symbiotic relationship between the administration of this probiotic strain and the host.

Another relevant property attributed to CIDCA 133 is its ability to inhibit enteropathogenic and other probiotic bacteria, an effect previously reported by Kociubinski et al. [22] and Hugo et al. [24] for other pathogens. The authors observed inhibition of CIDCA 133 against food spoilage and pathogenic bacteria *B. subtilis*, *B. cereus*, *P. aeruginosa*, and enterohemorrhagic *E. coli* O157:H7, and attributed all above inhibitory effects to the probiotic strain’s capacity to produce organic compounds, such as lactate.

The inhibitory effects of probiotics against pathogenic bacteria are also related to the production of bacteriocins. This property, as previously demonstrated by Oliveira et al. [106], showed that *L. rhamnosus* L156.4 inhibits the growth of pathogenic bacteria and other *Lactobacillus* by both the production of organic acids present in the strain supernatant and to the antibacterial activity of the bacteriocin enterocin A, whose gene was identified in its genome through BAGEL3 web server [106]. These findings support the present study results due to identifying the gene encoding the bacteriocins helveticin J and enterolysin A, and CIDCA 133′s ability to inhibit acid-resistance bacteria with a probiotic profile, such as *L. delbrueckii* CNRZ327 and *L. paracasei* BL23.

In conclusion, the genome-scale analysis of health-promoting probiotic CIDCA 133 elucidated many important functional roles of this strain. CIDCA 133 showed a broader repertoire of genes involved with molecular mechanisms related to its interaction with host, survival, adaptation, and immunostimulatory ability. The molecular bases attributed to the anti-inflammatory profile of CIDCA 133 can be associated with secreted and membrane/exposed to surface proteins. This is the first probiogenomics study of CIDCA 133, validated with in vitro and in vivo experiments, reinforcing that this strain is a highly effective probiotic, providing valuable benefits to the host.

## Figures and Tables

**Figure 1 microorganisms-09-00829-f001:**
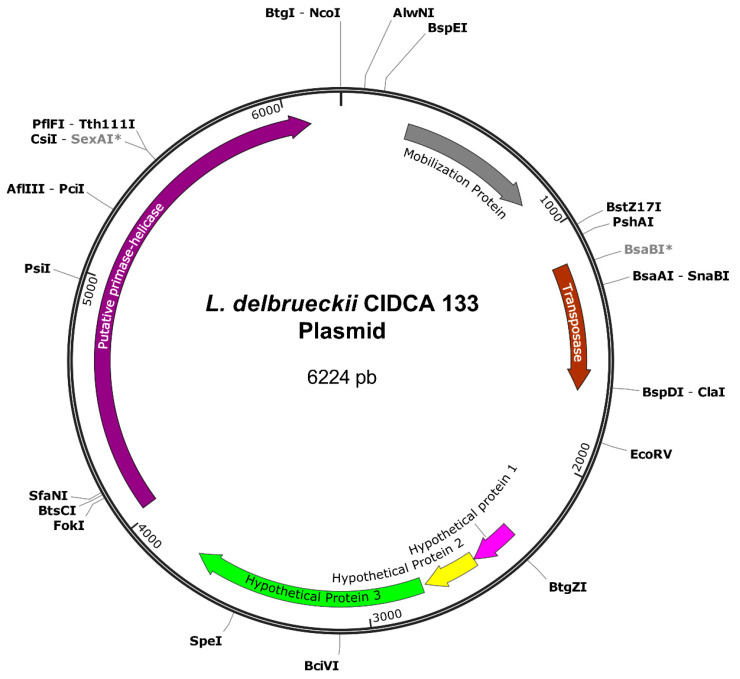
Schematic representation of the plasmid circular map present in *Lactobacillus delbrueckii* CIDCA 133 and restriction enzyme cut sites. Asterisk (*) indicates site blocked by methylation.

**Figure 2 microorganisms-09-00829-f002:**
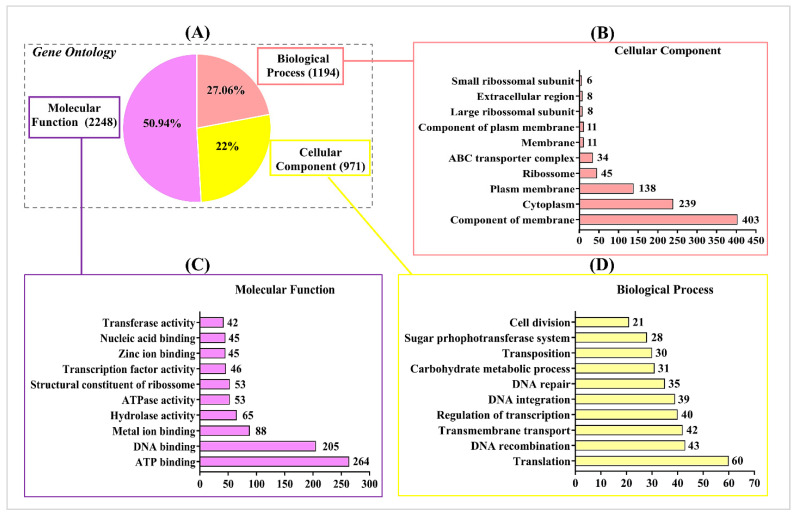
Distribution of *Lactobacillus delbrueckii* subsp. *lactis* CIDCA 133 (CIDCA 133) genes among the Gene Ontology (GO) categories predicted in Gene Ontology Functional Enrichment Annotation Tool (GO FEAT) (**A**). The genes were distributed in Cellular component (**B**), Molecular function (**C**), and Biological process (**D**) categories.

**Figure 3 microorganisms-09-00829-f003:**
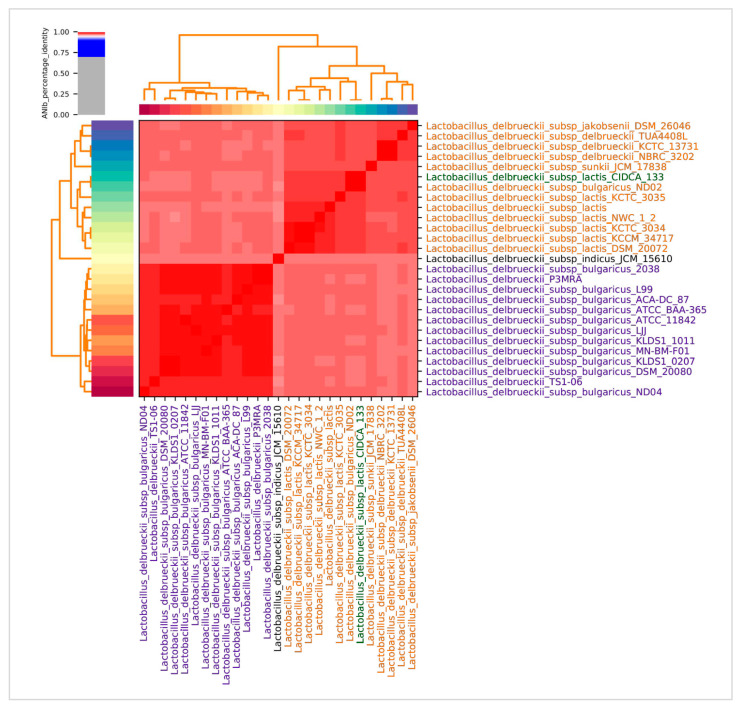
Heatmap representing the degree of similarity between *Lactobacillus delbrueckii* genomes based on the Average Nucleotide Identity (ANI). Red = more similar, light red = less similar.

**Figure 4 microorganisms-09-00829-f004:**
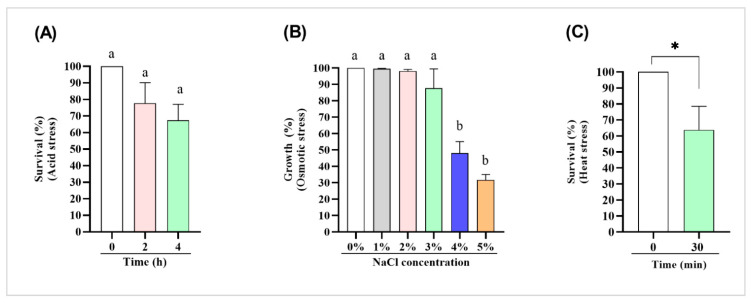
*L. delbrueckii* CIDCA 133 tolerates different stressors. (**A**) Survival percentage in acid stress (0.3% pepsin solution pH 3.0). (**B**) The growth rate in different concentrations of NaCl (1–5%). (**C**) Survival percentage in heat stress. Different letters and * indicate statistically significant differences (*p <* 0.05) by ANOVA followed by Tukey’s post hoc test (acid and osmotic stress) and Student’s *t*-test (thermal stress).

**Figure 5 microorganisms-09-00829-f005:**
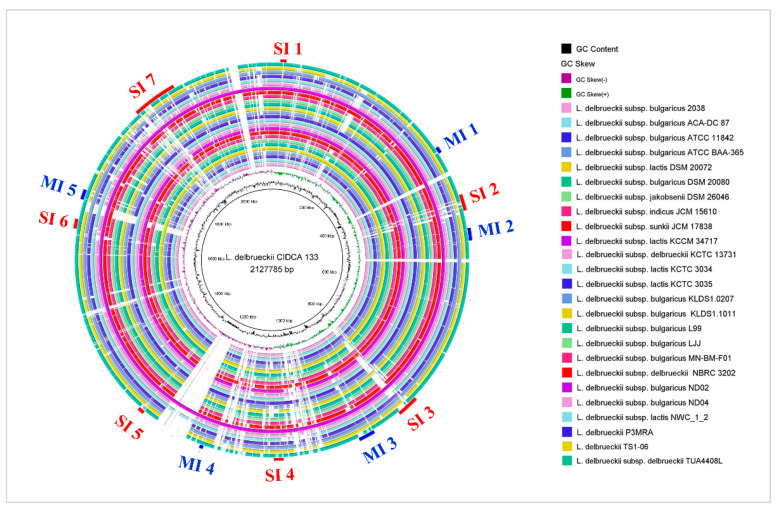
Circular schematic representation of Metabolic (MI) and Symbiotic (SI) islands predicted with GIPSy software in CIDCA 133 genome and its comparison with others *L. delbrueckii* complete genomes. Each ring of the circle corresponds to a specific *L. delbrueckii* whole genome, represented by different colors in the legend (right).

**Figure 6 microorganisms-09-00829-f006:**
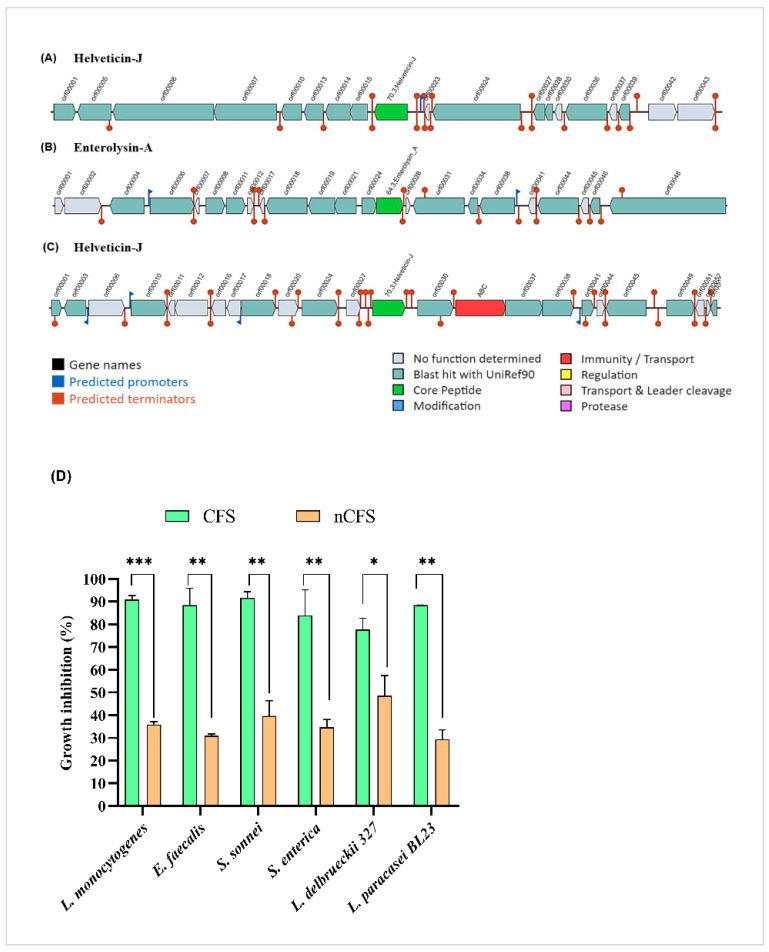
Schematic representation of bacteriocin gene operon (**A**–**C**) present in *L. delbrueckii* CIDCA 133 genome predicted by BAGEL4. (**D**) Antibacterial activity spectrum of bacterial supernatants from *L. delbrueckii* CIDCA 133 against pathogenic and non-pathogenic bacteria. Asterisk indicates statistical difference by Student’s *t-*test: * *p* < 0.05, ** *p* < 0.01, *** *p* < 0.001.

**Figure 7 microorganisms-09-00829-f007:**
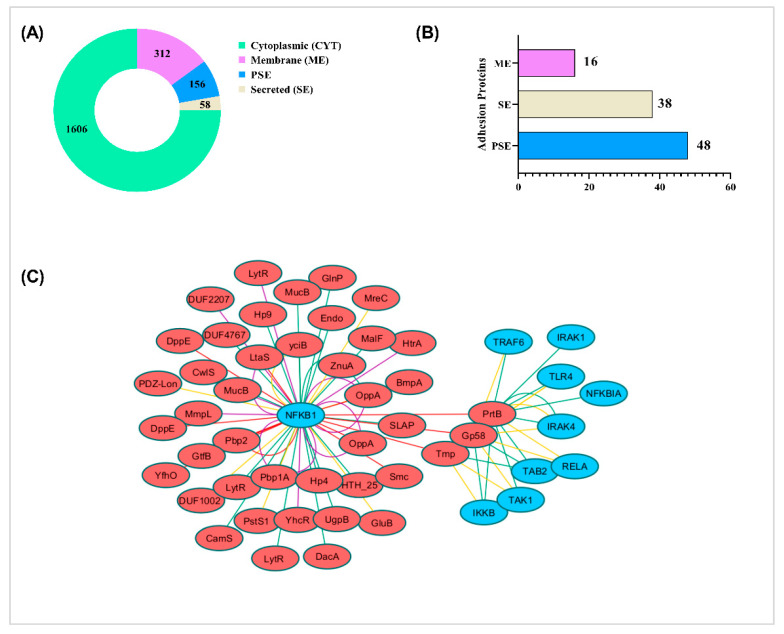
Distribution of subcellular localization of CIDCA 133 proteins predicted by SurfG+ software (**A**). Several proteins with high probably adhesion capacity predicted by Vaxign software (**B**). Protein–protein interaction network mapping to the KEGG toll-like receptor/nuclear factor κappa B (TLR/NF-κB) signaling pathway. Red circle nodes represent bacteria proteins, and blue circle nodes represent human proteins. The strongest associations are represented with different colors line. Interaction score: 0.97 (yellow line), 0.98 (green line), 0.99 (purple line), 1.0 (red line) (**C**).

**Figure 8 microorganisms-09-00829-f008:**
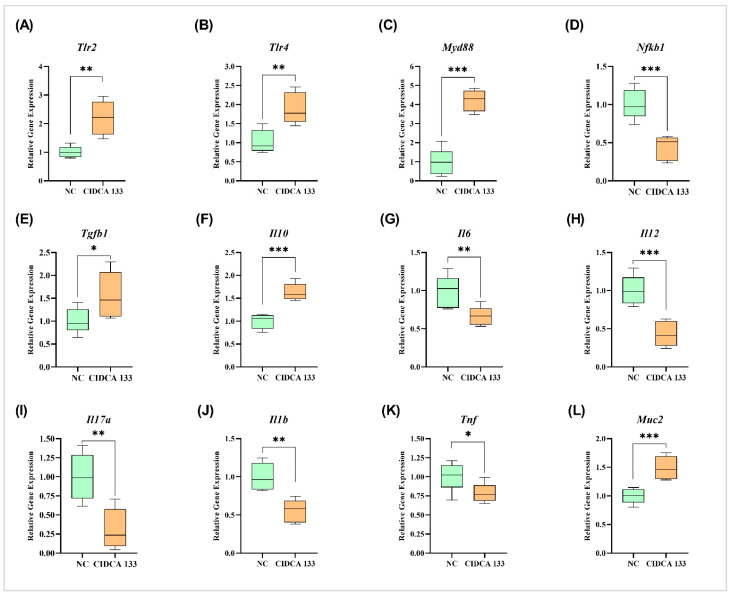
Relative gene expression of (**A**–**L**) *Tlr2*, *Tlr4*, *Myd88*, *Nfkb1*, *Tgfb1*, *Il10*, *Il6*, *Il12*, *Il17a*, *Il1b*, *Tnf*, and *Muc2* in the ileum section of animals that received oral administration of *L. delbrueckii* CIDCA 133 for 13 consecutive days. Asterisk indicates statistical difference by Student’s *t*-test: * *p* < 0.05, ** *p* < 0.01, *** *p* < 0.001.

**Table 1 microorganisms-09-00829-t001:** Complete genomes of *Lactobacillus delbrueckii* strains obtained from NCBI used in comparative analysis.

Nº	Bacteria Strain	Genome Access	Size (Mb)	GC%
1	*L. delbrueckii* P3MRA	NZ_CP045604.1	1.87	49.70
2	*L. delbrueckii* TS1-06	NZ_CP046390.1	1.85	49.80
3	*L. delbrueckii* subsp. *bulgaricus* LJJ	NZ_CP049052.1	1.89	49.50
4	*L. delbrueckii* subsp. *bulgaricus* KLDS1.1011	NZ_CP041280.1	1.89	49.80
5	*L. delbrueckii* subsp. *bulgaricus* MN-BM-F01	NZ_CP013610.1	1.88	49.70
6	*L. delbrueckii* subsp. *bulgaricus* KLDS1.0207	NZ_CP032451.1	1.87	49.80
7	*L. delbrueckii* subsp. *bulgaricu*s DSM 20080	NZ_CP019120.1	1.87	49.80
8	*L. delbrueckii* subsp. *bulgaricus* ND04	NZ_CP016393.1	1.86	49.60
9	*L. delbrueckii* subsp. *bulgaricus* ACA-DC 87	NZ_LT899687.1	1.86	49.80
10	*L. delbrueckii* subsp. *bulgaricus* L99	NZ_CP017235.1	1.85	49.70
11	*L. delbrueckii* subsp. *bulgaricus* 2038	NC_017469.1	1.87	49.70
12	*L. delbrueckii* subsp. *bulgaricus* ATCC 11842	NC_008054.1	1.86	49.70
13	*L. delbrueckii* subsp. *bulgaricus* ATCC BAA-365	NC_008529.1	1.86	49.70
14	*L. delbrueckii* subsp*. bulgaricus* ND02	NC_014727.1	2.13	49.59
15	*L. delbrueckii* subsp. *delbrueckii* NBRC 3202	NZ_AP019750.1	1.91	50.10
16	*L. delbrueckii* subsp. *delbrueckii* TUA4408L	NZ_CP021136.1	2.01	49.90
17	*L. delbrueckii* subsp. *delbrueckii* KCTC 13731	NZ_CP018216.1	1.91	50.00
18	*L. delbrueckii* subsp. *indicus* JCM 15610	NZ_CP018614.1	2.02	49.37
19	*L. delbrueckii* subsp. *jakobsenii* DSM 26046	NZ_CP018218.1	1.89	50.10
20	*L. delbrueckii* subsp. *lactis* KCCM 34717	NZ_CP018215.1	2.26	49.10
21	*L. delbrueckii* subsp. *lactis* KCTC 3034	NZ_CP023139.1	2.24	49.00
22	*L. delbrueckii* subsp. *lactis*1	NZ_LS991409.1	2.05	49.60
23	*L. delbrueckii* subsp. *lactis* KCTC 3035	NZ_CP018156.1	1.97	50.00
24	*L. delbrueckii* subsp. *lactis* NWC_1_2	CP029250.1	2.26	48.58
25	*L. delbrueckii* subsp. *lactis* DSM 20072	NZ_CP022988.1	2.17	49.00
26	*L. delbrueckii* subsp. *sunkii* JCM 17838	NZ_CP018217.1	2.00	50.10

**Table 2 microorganisms-09-00829-t002:** Quantitative Polymerase Chain Reaction (qPCR) primers used in this study.

Gene	Primer Forward	Primer Reverse	Amplicon Size (bp)	Reference
*Actb*	GCTGAGAGGGAAATCGTGCGTG	CCAGGGAGGAAGAGGATGCGG	100	[58]
*Gapdh*	TCACCACCATGGAGAAGGC	GCTAAGCAGTTGGTGGTGCA	168	[59]
*Il6*	GAGGATACCACTCCCAACAGACC	AAGTGCATCATCGTTGTTCATACA	141	[59]
*Il10*	GGTTGCCAAGCCTTATCGGA	ACCTGCTCCACTGCCTTGCT	191	[59]
*Il12p40*	GGAAGCACGGCAGCAGAATA	AACTTGAGGGAGAAGTAGGAATGG	180	[59]
*Tnf*	ACGTGGAACTGGCAGAAGAG	CTCCTCCACTTGGTGGTTTG	236	[60]
*Il1b*	CTCCATGAGCTTTGTACAAGG	TGCTGATGTACCAGTTGGGG	245	[60]
*Il17a*	GCTCCAGAAGGCCCTCAGA	AGCTTTCCCTCCGCATTGA	142	[59]
*Tgfb1*	TGACGTCACTGGAGTTGTACGG	GGTTCATGTCATGGATGGTGC	170	[59]
*Muc2*	GATGGCACCTACCTCGTTGT	GTCCTGGCACTTGTTGGAAT	246	[58]
*Myd88*	ATCGCTGTTCTTGAACCCTCG	CTCACGGTCTAACAAGGCCAG	199	[61]
*Tlr2*	ACAATAGAGGGAGACGCCTTT	AGTGTCTGGTAAGGATTTCCCAT	149	[61]
*Tlr4*	ATGGCATGGCTTACACCACC	GAGGCCATTTTTGTCTCCACA	129	[61]
*Nfkb1* (p105)	GTGGAGGCATGTTCGGTAGTG	TCTTGGCACAATCTTTAGGGC	195	[62]

**Table 3 microorganisms-09-00829-t003:** Gene’s prediction involved with stress tolerance of *Lactobacillus delbrueckii* CIDCA 133.

Locus Tag	Gene	Protein	Stress Condition
HR078_02445	*Odcl*	Ornithine decarboxylase	Acid
HR078_03205	*atpD*	F0F1-ATP synthase subunit beta	Acid
HR078_03195	*atpA*	F0F1- ATP synthase subunit alpha	Acid
HR078_03180	*atpE*	F0F1-ATP synthase subunit C	Acid
HR078_03185	*atpF*	F0F1-ATP synthase subunit B	Acid
HR078_03210	*atpC*	F0F1-ATP synthase epsilon	Acid
HR078_03190	*atpH*	F0F1-ATP synthase delta	Acid
HR078_03200	*atpG*	F0F1-ATP synthase gamma	Acid
HR078_03175	*atpB*	F0F1-ATP synthase subunit A	Acid
HR078_03560	*clpX*	ATP-dependent ClpX protease	Acid
HR078_00335	*ark*	Aldo/keto reductase	Osmotic
HR078_09455	*glpF*	Aquaporin family protein	Osmotic
HR078_10525	*nagB*	Glucosamine-6-phosphate deaminase	Biliar
HR078_01470	*pyrG*	CTP synthase	Biliar
HR078_09705	*pepF*	Oligoendopeptidase F	Biliar
HR078_04350	*cbh*	Choloylglycine hydrolase family protein	Biliar
HR078_07785	*groel*	chaperonin GroEL	Heat
HR078_06405	*hcrA*	Heat-inducible transcription repressor HrcA	Heat
HR078_06390	*dnaJ*	Molecular chaperone DnaJ	Acid, Biliar, Osmotic, Heat
HR078_06395	*dnaK*	Molecular chaperone DnaK	Acid, Biliar, Osmotic, Heat
HR078_06400	*grpE*	Nucleotide exchange factor GrpE	Acid, Biliar, Osmotic, Heat
HR078_00560	*YyclC*	Two-component system regulatory protein	Acid, Biliar, Osmotic, Heat
HR078_08640	*nhaC*	Na+/H+ antiporter NhaC	Acid, Osmotic
HR078_06090	*clpP*	Clp protease ClpP	Acid, Biliar
HR078_06270	*clpE*	AAA family ATPase	Acid, Biliar
HR078_06320	*eno*	Phosphopyruvate hydratase	Acid, Biliar
HR078_00380	*luXs*	S-ribosylhomocysteine lyase	Osmotic, Biliar

## Data Availability

Complete genome sequences used for these analyses are available from the GenBank, NCBI.

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
