# Peer review of "Probiogenomics of Lactobacillus delbrueckii subsp. lactis CIDCA 133: In Silico, In Vitro, and In Vivo Approaches"

_microorganisms, 2021, doi:10.3390/microorganisms9040829_

Round 1

Reviewer 1 Report

Lactobacillus strains are recognized as probiotics with immunomodulatory and anti-inflammatory abilities, efficient in treatment of various diseases, and finding applications in food industry. This manuscript presents a comprehensive in silico, in vitro and in vivo analysis of CIDCA 133 – a new strain of Lactobacillus delbrueckii subfamily – including estimation of its tolerance for acid, bile, osmotic and heat stress. The methods used are described in a clear and concise fashion and supported by references; the results are appropriately summarized in the Discussion section. While the scope of the present paper is quite broad, the text is well-structured and easy to follow.

In my opinion, this paper will be of use for specialists interested in probiotics and their applications and is suitable for publication in Microorganisms.

Typos:

Line 517: “it” is redundant.

Line 557: promoting -> promotes.

Line 578: “… through of the BAGEL3” – please correct.

Author Response

All comments have been valuable and very helpful for revising and improving our paper (Please see the attachment).

Reviewer 2 Report

The manuscript submitted by  Lima de Jesus et al provide many informations about genetics and probiotic attributes of the Lactobacillus delbrueckii subsp. lactis CIDCA 133 strain. This work combine several approaches including in silico analysis of the genome and in-vitro tests strenghtening knowledges of this probiotic microorganism. 
The amount and the quality of the data gathered and displayed in the manuscript allow to have a comprehensive picture of the metabolic and beneficial attributes of this strain. However some remarks and questions listed bellow needs to be precised

Line 24-25: Please rephrase as following « Phenotypic assays demonstrated that CIDCA 133 strain could survive to acidic, osmotic, and thermic stresses”

Line 26: Replace “plays” by “shows”

Line 30: Please rephrase the following sentence “these findings describe”

Line 71-72: Please check this sentence, something seems to be missed here!

Line 82: “de Man, Rogosa and Sharpe”

Line 173: Modify “ability to resist to artificially simulated gastric juice”

Line 185: Any serial dilution applied here? If yes please specify it in this part.

Line 213-214: This experiment seems to be a kind of a killing curves assay! Why authors didn’t investigate the antibacterial activity using the agar diffusion test method, which is generally used in this case? This method allows to unveil the antagonistic activity and quantify it using arbitrary units!

Line 228-229: Which volume of MRS broth and CIDCA 133 was given to the mice daily? A control group “not fed with MRS” is missing here!

Table 2: Please add a column with the expected sizes of the amplicons

Figure 3: The strain CIDCA133 should be highlighted. The names of the strains belonging to the two different subgroup should be in different colors to distinguish them.

Line 457: “BAL species”?

Line 458: Modify as following “this fact, with only four plasmid sequences deposited” and give accession numbers of these sequences.

Line 531: Replace “corroborates” by “is in accordance”

Line 534: Which kind of studies?

Line 585-586: Please rephrase this sentence

Author Response

(The authors gave the same response as above.)
